# The Role of Nutrition in Immune-Mediated, Inflammatory Skin Disease: A Narrative Review

**DOI:** 10.3390/nu14030591

**Published:** 2022-01-29

**Authors:** Federico Diotallevi, Anna Campanati, Emanuela Martina, Giulia Radi, Matteo Paolinelli, Andrea Marani, Elisa Molinelli, Matteo Candelora, Marina Taus, Tiziana Galeazzi, Albano Nicolai, Annamaria Offidani

**Affiliations:** 1Dermatological Clinic, Department of Clinical and Molecular Sciences, Polytechnic Marche University, 60100 Ancona, Italy; federico.diotallevi@hotmail.it (F.D.); anna.campanati@gmail.com (A.C.); radigiu1@gmail.com (G.R.); matteopaolinelli.an@gmail.com (M.P.); andreamarani.med@yahoo.com (A.M.); molinelli.elisa@gmail.com (E.M.); matteocandelora@gmail.com (M.C.); a.offidani@ospedaliriuniti.marche.it (A.O.); 2Dietetic Unit and Clinical Nutrition, United Hospitals of Ancona, 60100 Ancona, Italy; m.taus@ospedaliriuniti.marche.it (M.T.); a.nicolai@ospedaliriuniti.marche.it (A.N.); 3Department of Pediatrics, Marche Polytechnic University, 60100 Ancona, Italy; t.galeazzi@univpm.it

**Keywords:** nutrition, diet, immune-mediated skin diseases, psoriasis, atopic dermatitis, suppurative hidradenitis, bullous diseases, vitiligo, alopecia areata

## Abstract

Immune-mediated inflammatory skin diseases are characterized by a complex multifactorial etiology, in which genetic and environmental factors interact both in genesis and development of the disease. Nutrition is a complex and fascinating scenario, whose pivotal role in induction, exacerbation, or amelioration of several human diseases has already been well documented. However, owing to the complexity of immune-mediated skin disease clinical course and breadth and variability of human nutrition, their correlation still remains an open debate in literature. It is therefore important for dermatologists to be aware about the scientific basis linking nutrition to inflammatory skin diseases such as psoriasis, atopic dermatitis, hidradenitis suppurativa, bullous diseases, vitiligo, and alopecia areata, and whether changes in diet can influence the clinical course of these diseases. The purpose of this narrative review is to address the role of nutrition in immune-mediated inflammatory skin diseases, in light of the most recent and validate knowledge on this topic. Moreover, whether specific dietary modifications could provide meaningful implementation in planning a therapeutic strategy for patients is evaluated, in accordance with regenerative medicine precepts, a healing-oriented medicine that considers the whole person, including all aspects of the lifestyle.

## 1. Introduction

Immune-mediated and inflammatory skin diseases (IMID), including psoriasis, atopic dermatitis, suppurative hidradenitis, bullous diseases, vitiligo, and alopecia areata, although very clinically different from each other, share a chronic inflammatory background of the skin. Most of the patients with IMID give their dietary habits a significant role in disease course. They often ask dermatologists about the role of diet in inducing, exacerbating, or managing skin disorders, and not infrequently they turn to alternative sources of information to obtain answers, as they report receiving inadequate information from clinicians. It is, therefore, crucial for dermatologists to be aware of the growing body of literature concerning nutrition and immune-mediated skin disease to make patients appropriately informed about potential benefits of specific dietary interventions.

Previous studies have already demonstrated the potential role of diet on the risk of exacerbating IMID [1]. Although epidemiological studies point out some dietary factors as possible inducers of IMID, it is not clear as to whether and how diet could influence IMID development and which foods may have a crucial role in diseases progression and flares [2]. However, it has already known that several components of foodstuff have been indicated as potential triggers, including processed foods and their additives, and low levels of nutrients have also been associated with increased inflammation (e.g., vitamin D serum levels).

Although apparently IMID may be very different clinically, these diseases share common physiopathological pathways and often the same targeted treatment regimens.

Hence, analyzing data from the literature could improve our knowledge on the impact that dietary modifications may have on the control of the disease and associated comorbidities. Evidence supporting the adjuvant role of various diets in managing IMID is summarized in this review.

## 2. Materials and Methods

This narrative review was based on the general approach developed biomedical narrative review construction, which consists of four key steps: (1) identify keywords; (2) conducting research; (3) review abstract and articles; (4) document results [3,4].

### 2.1. Identify Keywords

To identify keywords, we used a brainstorming approach, involving the entire research group.

The keywords selected were “nutrition AND immune mediated skin diseases”, “nutrition AND psoriasis”, “nutrition AND suppurative hidradenitis”, “nutrition AND bullous skin diseases”, “nutrition AND atopic dermatitis”, “nutrition AND vitiligo”, “nutrition AND alopecia areata”. The same associations were made with the term “diet”.

### 2.2. Conduct Research

We performed a worldwide review of studies reporting nutrition in immune mediated skin diseases using 3 electronic medical databases—PubMed, EMBASE, and Web of Science. The search terms were selected to identify studies describing the role of nutrition in IMID. All selected databases were searched from their respective inception in the last 10 years. In addition, we searched by hand the reference lists of other relevant articles on nutrition involvement in immune-mediated skin diseases. Additionally, government reports, as well as gray literature available on nutrition and IMID, were searched.

In this first phase, 202 records were identified from the selected database. Records after duplicates were removed were 181 in number. Among the selected records, none were marked as ineligible by automation tools. No other reason for removing records was identified.

### 2.3. Review Abstract and Article

The selection of the relevant studies took place in three steps. In the first step, six researchers (G.R., M.P., A.M., E.M., T.G.) made independent selections of articles on the basis of title. Any disagreements were resolved by consulting a senior investigator (A.O.). The second step was to evaluate the abstracts. At least two members of the research team (A.C., F.D., M.T.) independently evaluated each abstract and any discrepancies were additionally resolved through unanimous consent. Seventy-five papers were excluded, and 106 were assessed for full text analysis. Among them, 6 manuscripts were not retrieved. The third phase consisted of critical appraisal of the full text of the 100 selected articles. A final sample of 45 studies was included in qualitative synthesis. Inclusion criteria were the following: studies reporting on nutrition in IMID, studies published in the English language, abstract available, and no restrictions in study design, and randomized controlled trials, case–control studies, cross-sectional studies, and case series were included. Exclusion criteria were as follows: review articles (10 reports), case reports (30 reports), and reports in languages other than English (5 reports).

### 2.4. Document Results

All sources with similar data/level of evidence were analyzed, collected, and grouped. The main text was structured into subsections. New evidence-based points were summarized, and major points for future research and practice were defined.

## 3. Results

Our search identified 181 records after removing duplicates. After scanning the titles and abstracts, we excluded 75 citations, and 106 were assessed for full-text eligibility. After examining the full text, we considered 45 case–control studies, case series studies, randomized controlled trials, and meta-analyses studies eligible and included them in this study.

The data found have showed that nutrition has a conditioning role in many inflammatory and immune-mediated diseases of the skin, namely, in psoriasis, atopic dermatitis, hidradenitis suppurativa, bullous diseases, vitiligo, and alopecia areata.

### 3.1. Psoriasis

Psoriasis is a chronic inflammatory disease with a prevalence of about 2–3% in the general population [5]. It is a multifactorial disease in which genetic and environmental factors contribute to the systemic inflammatory state of the disease [6]. The systemic inflammatory state, characterized by the increased expression of tumor necrosis factor alfa (TNF-alfa)/interleukin-23 (IL-23)/IL-17 axes [7], results in the skin being involved together with other organs as well [8,9,10]. It is known that several pathological conditions can be associated with psoriasis, including psoriatic arthritis, gastrointestinal disease, obesity, and metabolic disease (MetS), which increased cardiovascular risk in psoriatic patients and mood disorders [11]. Obesity, being associated with an increase in visceral fat, is characterized by low-grade inflammatory status with production of several of cytokines and chemokines such as beta-defensins 2/3, CXCL8/10, and CCL20, which promote the development of psoriasis [12]. Consequently, it is reasonable to assume that a diet resulting in weight loss and decrease in visceral fat may be helpful in patients with psoriasis. However, in the scientific community, there has been uncertainty regarding the role of dietary interventions in the treatment of psoriasis.

In 2018, on the basis of a systematic review of the literature, the Medical Board of the National Psoriasis Foundation released dietary recommendations for adults with psoriasis [13]. The authors, through the analysis of 55 studies of 77,557 patients, including 4534 with psoriasis, found high-quality evidence to support weight reduction with a hypocaloric diet as an adjunct to standard medical therapy for overweight or obese adults (body mass index (BMI) ≥ 25) with psoriasis. In particular, it refers to a low-calorie diet with a daily calorie intake ranging from 800 to 1400 kcal, lasting from 16 weeks to 6 months with a consequent improvement in disease severity, dermatology-life quality index (DLQI), and weight loss [14,15,16,17,18]. Diet seems to be especially useful in patients taking either standard therapy, such as topical therapies, phototherapy, cyclosporine, methotrexate, acitretin, or biological therapy [14,15,19,20]. However, in two studies in which the diet was combined with systemic treatment of methotrexate and cyclosporine, after treatment discontinuation and maintenance of the hypocaloric diet, there was no maintenance of disease clearance. It would therefore appear that diet is not sufficient to control the disease [15,21]. Additional studies are necessary to explore the effects of specific dietary patterns on psoriasis. A French web-based questionnaire cohort study found an inverse association between psoriasis severity and degree of adherence to the Mediterranean diet (a diet high in fruits, vegetables, legumes, cereals, bread, fish, fruit, nuts, and extra-virgin olive oil) [22]. However, data are insufficient to confirm a beneficial effect of this diet.

The gluten-free diet appears to be of definite benefit in patients with psoriasis and confirmed celiac disease [13]. Several studies show that a gluten-free diet not only improves gastrointestinal symptoms but also reduces the severity of the disease [23,24,25]. In addition, the Medical Board of the National Psoriasis Foundation suggested a three-month trial of a gluten-free diet as an adjunct to standard medical therapy in adults with psoriasis who test positive for serologic markers of gluten sensitivity [13]. Universal screening of individuals with psoriasis for gluten sensitivity was discouraged by the American Collage of Gastroenterology in favor of limiting screening to individuals with a first-degree relative with celiac disease or active gastrointestinal symptoms [26].

Some data from literature have already demonstrated both the metabolic implications in psoriasis and the beneficial role of a low-calorie restricted diet in disease improvement and response to systemic therapies. However, reliable data on nutritional supplementation, which has prompted doctors to recommend supplements such as fish oil, vitamin D, selenium, and vitamin B12 for psoriasis, is still absent [13,27,28,29,30].

### 3.2. Atopic Dermatitis

Atopic dermatitis (AD) is a multifactorial inflammatory skin disorder that affects approximately 5–20% of children and 1–3% of adults [31,32]. It is characterized by the presence of eczematous lesions and intense itching, with a strong impact on the quality of life of patients and their caregivers.

The T2-mediated pathways that support atopic dermatitis are also responsible for the presence of other atopic type 2 comorbidities such as allergic rhino-conjunctivitis, allergic asthma, and food allergies that are often associated with atopic dermatitis. Nutrition and, specifically, the role of certain foods as potential triggers of atopic dermatitis has long been debated in the literature, as well as the attempt to attribute to diet a therapeutic effect on atopic dermatitis.

The Finch et al. [33] study reviewed the impact of nutrition in AD, from maternal nutrition in pregnancy through adulthood. Maternal dietary restrictions during pregnancy or lactation have no effect on incidence or severity of AD. It appears that reduced maternal egg intake during pregnancy reduces the incidence of egg allergy in children tested by prick testing, but this does not translate into reduced incidence of AD. In children at high risk for atopy (defined as having a first-degree relative with atopic dermatitis, asthma, or allergies) whose mothers received prophylactic administration of prenatal Lactobacillus GG, the frequency of AD was reduced by half compared with the placebo group (*p* = 0.008). Breastfeeding also appears to have a protective role in children at high risk for AD.

Data regarding the utility of delayed introduction of solid foods to prevent AD are discordant. Hydrolyzed formulas (extensively hydrolyzed casein and partially hydrolyzed whey) resulted in a significantly reduced incidence of AD at 1 year compared with children given cow’s milk formula; however, use of hydrolyzed formulas offered no advantage over exclusive breastfeeding.

Dietary restrictions do not seem to improve atopic dermatitis, either in adults or in children for whom dietary restriction can often cause growth alterations. Very often, restrictions are initiatives taken by patients without medical advice. The study by Low et al. [34] showed that in a population of 150 children with AD aged 12 to 36 months, about 60% were subject to food restriction, defined as the deprivation of three or more foods from the diet. This measure was not prescribed by the doctor but taken on by the parents of the children on their own initiative. The foods avoided were mainly shellfish (62.7%); nuts (55.3%); egg (50.0%); dairy, including all milk products, including cow, goat, and camel milk (29.3%) and cow’s milk (28.7%). Earlier AD onset, more severe disease, and lower maternal education level seem to be the predisposing factors for food restriction practice. The authors evaluated key growth parameters of height, weight, head circumference, and mid-upper arm circumference (MUAC) by comparing them to the World Health Organization (WHO) Child Growth Standards 2006 charts. It was observed that children with AD have lower than average growth. Children with dietary restrictions have intakes of calories, protein, carbohydrates, fat, riboflavin, vitamin B12, and phosphorus that met or exceeded the recommended nutrient intake(RNI) was significantly lower than non-restricted. Growth parameters were also lower: mean height, weight, head circumference, and MUAC of children with dietary restrictions were significantly lower than those without dietary restrictions. Younger age of disease onset and more severe disease were significantly associated with dietary restriction. AD severity was an independent risk factor affecting height and weight, regardless of dietary restriction. The absence of a control group due to the cross-sectional study design makes it difficult to establish a causal relationship between dietary restrictions and growth. In this study, data collection on eating habits was conducted on an anamnestic basis, and no laboratory tests were performed to check for food allergies. It remains difficult to establish whether the intake of certain foods can trigger atopic dermatitis and to select in which cases there may be an underlying food allergy. According to some data obtained from the literature [35], the possible mechanisms by which food intake can exacerbate AD are twofold: IgE-mediated and non-IgE-mediated. IgE-mediated forms are rapid onset and can be verified by skin prick tests and a search for specific IgE in blood. In these cases, elimination of food from the diet reduces the severity of atopic dermatitis. Non-IgE-mediated reactions arise hours or days after ingestion of a food and are often difficult to diagnose. Deprivation of suspected foods in individuals with AD for whom there is no diagnosis of food allergy does not benefit the dermatitis and should be avoided. The review by Guibas et al. [36] also highlighted the same critical issues reported in the literature on AD and food allergies. Although some patients report exacerbation of AD after intake of certain foods, nutritional restrictive measures often taken on their own initiative by patients are not effective. In most cases, it is difficult to understand the pathophysiological mechanism that can induce a worsening of AD (type I or type IV) and to establish a causal link between the foods or rather the additives contained in food. There are also histamine-releasing foods able to generate itching without there being an actual underlying allergic reaction. In addition, the systems of investigation are not univocal: there are diagnostic tests performed in pharmacies outside a medical context of dubious interpretation, and thus often avoidance of certain foods is undertaken without real necessity. The authors concluded that in adults, in-depth investigations should be performed under medical guidance, as well as in patients with recalcitrant AD when the suspicion of atopic predisposition is high. Only in severe forms of AD is the search for food allergies by prick test or radioallergosorbent assays is recommended.

Several studies have been conducted on subjects with AD and IgE-mediated food allergies documented by skin prick test (SPT) and IgE detection in the blood.

Marie-Helene et al. [37] conducted a 3 year prospective study on a pediatric population of 100 children that SPT-positive and IgE-positive for foods, specifically egg in 67%, peanut in 54%, milk in 30%, sea shells in 26.9%, wheat flour in 16.8%, fish in 11.2%, soy in 8.9%, and mustard in 4.5%. During the observation period, patients avoided foods they were allergic to. The authors observed reduction in SCORing Atopic Dermatitis (SCORAD) values and mean topical steroid consumption, associated with significant reduction in food-specific IgE levels. Reintroduction was possible within three years in 90% of patients with egg allergy, 95% for milk, 100% for wheat flour, and 50% for soybeans; the mean age for regression of food allergy under an eviction diet was 18 months for milk, 3 years for flour and eggs, and none for seafood and peanuts. They suggest that in pediatric patients with a definite diagnosis of food allergy, dietary restriction of that food may reduce the severity of dermatitis and the need for topical treatment.

Olendzka et al. [38] noticed some difference in the efficacy of elimination diet among a pediatric population according by age. They examined 140 children who were tested by skin prick tests and measurement of specific IgE in blood. An elimination diet in infants and children under age three was the most effective. In older children with atopic dermatitis, the elimination diet should be combined with antihistamine medications and local skin treatment.

Finally, Rokaite et al. [39] studied 154 children aged 6 months to 18 years whose food allergens were determined by allergy skin tests (prick skin and patch). The authors evaluated the number of AD flares during one year of a specific restriction diet. Each patient was given a personalized diet on the basis of the tests performed with elimination of foods they were allergic to and replacement with other foods. After one year, subjects who were able to follow the recommended diet suffered more rarely from severe allergic rash problems during a 12 month period (*p* = 0.01) and had to take less allergy medication compared to children who did not follow the dietary recommendations (*p* = 0.001). After comparison of skin patch test results (before dietary treatment and after a 1 year period), it was determined that only skin patch tests against buckwheat, oats, beef, and cocoa did not change in a statistically significant manner.

Dietary modifications in individuals with AD, beyond dietary restrictions, have also included consideration of nutritional supplementation with potential therapeutic effects on AD [40]. With regard to child feeding, there is insufficient evidence to recommend prebiotics in AD.

Although basic scientific research supports a mechanistic role for flavonoids in inflammatory disease, there is currently no evidence that flavonoids have any significant effect on AD [35]. The ω-3 and ω-6 oils, although seemingly ineffective in preventing disease, may have some benefit when used to decrease disease severity [33], as well as a vitamin supplementation of vitamin A and C in selected populations. In addition, dietary supplementation of docosahexaenoic acid (DHA) and fish oil, vitamin D, and vitamin E has shown improvement in symptoms in several randomized trials [35].

Kim et al. [41] conducted a study on the potential therapeutic role of vitamin D in the control of the disease. Through a literature review, they collected available data from observational studies and randomized controlled trials (RCTs) from Cochrane, Medline, and Embase databases. From 920 selected studies, 11 were included in the meta-analysis. The data showed that serum vitamin D values were lower in subjects with AD than in healthy controls, particularly in the pediatric population. After dietary vitamin D supplementation, SCORAD and Eczema Area and Severity Index (EASI) scores decreased, highlighting a therapeutic role of vitamin D in improving disease severity after dietary supplementation in both children and adults, although post-treatment vitamin D serum values were not reported.

In conclusion, it is possible that some foods lead to a worsening of AD. Avoidance behaviors should be adopted only in cases of food allergies that can be tested through SPK and IgE research on blood. Only subjects with severe or refractory forms of AD should investigate food allergies, following a diagnostic pathway under medical guidance. In children, non-specific food restriction could lead to growth alteration without benefit to AD. Limited evidence is currently available to support the beneficial effect of vitamin, fatty acid, flavonoid, and probiotic supplementation in subjects with AD.

### 3.3. Hidradenitis Suppurativa

Hidradenitis suppurativa (HS), also known as acne inversa or Verneuil’s disease, is a chronic inflammatory condition localized to the axillary, inguinal, perineal, gluteal, pubic, and mammary regions, i.e., areas rich in apocrine glands [42]. Its prevalence in the general population is 1% [43]. It is characterized by the presence of deep nodules, abscesses, fistulous tracts, and fibrotic outcomes [43]. The etiology of HS is multifactorial and includes genetic, environmental, and behavioral factors [44,45]. In particular, individuals with HS are more frequently overweight or obese. Obesity leads to increased intertriginous surface area and skin rubbing, increasing sweat production. Moreover, it determines hormonal modifications with relative increase of androgens implicated in the pathogenesis of HS [46,47].

Several studies have been carried out to outline the role of diet or nutrient supplementation in the course of the disease. A study with 242 participants showed that 75.6% of patients with HS had modified their eating habits, 30.9% of which reported an improvement in symptomatology after such modification [48]. Indeed, obesity leads to insulin resistance with increased insulin and IGF-1 and hyperandrogenism. This stimulates androgen receptors that increase intraductal keratinocyte production. Hyperkeratinization results in follicular occlusion, rupture, inflammation, and skin damage. This leads to exposure of free antigens to commensal bacteria, formation of fistulas, and scarring. It is therefore easy to understand how a reduction in weight with a consequent reduction in the consequences of obesity can result in an improvement in symptoms [49].

In addition, it has been seen that some foods such as dairy products and simple sugars in general determine a worsening of symptoms [50]. On the contrary, fruits, vegetables, chicken meat, and fish seem to improve the symptomatology itself. This appears to be due to the high content of micronutrients such as zinc and vitamin B12 in these foods, and indeed several systematic reviews indicate that their supplementation may improve the symptoms of HS. Zinc at a dosage of 90 mg/day in particular seems to be able to stimulate innate immunity in patients with otherwise deficient HS [51,52].

Dairy products and simple carbohydrates result in increased levels of insulin and insulin-like growth factor-1 (IGF-1), leading to activation of FOXO1, a transcription factor and regulatory protein that normally suppresses mTORC1 (mammalian target kinase of rapamycin complex 1). FOXO1 activation disinhibits mTORC1 signaling with cellular hyperproliferation of follicular epithelium and sebaceous glands. Furthermore, activation of FOXO1 by IGF-1 exposes androgen receptors within the pilosebaceous unit, potentiating the hyperproliferative effects. Therefore, activation of mTORC1 signaling and androgenic hyperactivity induces excessive lipogenesis of sebum, predisposing to a follicular occlusion event believed to trigger both acne and HS. Dairy products and brewer’s yeast seem to be specifically implicated in worsening HS through this mechanism [50,51,52].

Omega-3 fatty acids appear to play a role in reducing inflammation, and may therefore have a beneficial role in HS. Similarly, vitamin D deficiency has been found in patients with HS. In particular, the degree of insufficiency appears to correlate with the severity of the condition. It appears that vitamin D is able to reduce inflammatory nodules, probably by stimulating innate skin immunity, leading to clinical improvement of the pathology itself. However, there is still no firm evidence in this regard [53]. Fifty percent of patients who discontinue vitamin D supplementation relapse suggesting that it may have a more prophylactic role on relapses rather than curative. In general, vitamin D enhances innate immunity through expression of Toll-like-receptor type 2 (TLR-2) and peptides with antimicrobial activity. Vitamin D also influences keratinocyte proliferation, thus being able to prevent follicular obstruction typical of HS [53,54].

A significant association between HS and diabetes mellitus has been demonstrated. HS results in a major increase in various cytokines and inflammatory markers, particularly TNF-α. TNF-α can inhibit insulin signaling in particular phosphorylation of insulin type 1 receptor substrate. Altered regulation of this pathway has been associated with the development of insulin resistance and diabetes, and this is the possible mechanism by which HS is associated with diabetes mellitus(DM) [55]. In addition, HS is associated with an elevation of inflammatory indices such as PCR and leukocyte count higher than in other inflammatory dermatoses such as psoriasis as well as a reduced number of endothelial progenitor cells. All this has been associated with an altered glycemic control and altered function of beta-pancreatic cells, which could be a contributing factor to DM. There is also a possibility that hyperglycemia, insulin resistance, and DM predispose the patient to developing HS or exacerbates it. In fact, hyperglycemia is able to hyperactivate the mammalian target of rapamycin 1 (m-TOR1) pathway involved in keratinocyte proliferation, sebaceous glands, and lipogenesis. This may predispose the patient to develop the clinical features of HS such as follicular occlusion, fistula formation, and eventual scarring [55].

Therefore, further studies are needed to definitively demonstrate the usefulness of specific dietary regimens in the treatment of HS, as well as whether it is more effective to eliminate certain foods from the diet or avoid others. However, patients may benefit from a change in dietary regimen as early as possible as part of a broader treatment for HS.

### 3.4. Bullous Diseases

Immune-mediated bullous diseases represent a large and heterogeneous chapter in terms of clinical manifestations and pathophysiology. Therefore, the role of nutrition in these diseases is complex and varied. The discussion focuses mainly on dermatitis herpetiformis (DH), pemphigus, and bullous pemphigoid. They are acquired blistering diseases with DH and bullous pemphigoid belonging to the subepidermal group, with pemphigus to the intraepidermal group.

#### 3.4.1. Dermatitis Herpetiformis

DH is an extremely itchy, papulovesicular skin disease characterized by the presence of granular IgA deposits in the dermal papillae and/or granular deposits along the basement membrane with accentuation at the papillary tips [56]. DH is considered a cutaneous manifestation of coeliac disease (CD), although the exact causal mechanism is not known. The signs and symptoms of DH typically appear around 30 to 40 years of age, although all ages may be affected [57]. The classical clinical presentation consists of papules, vesicles, and urticarial plaques affecting, symmetrically, the extensor surfaces of the limbs, buttocks, the sacral region, neck, face, and scalp, while mucous membranes are usually spared [58]. As a specific manifestation of CD, a lifelong gluten-free diet, is the first-choice treatment of the disease. However, months or even years are needed before diet alone can control the dermatologic manifestations, especially itching. Thus, in the first phase of the treatment, several drugs can be used for variable periods of time to obtain a relief of the symptoms including dapsone or corticosteroids [59]. A scrupulous gluten-free diet (GFD) is the mainstay for the treatment of CD and related disorders. GFD is able to treat both the gastrointestinal and the cutaneous manifestations and is also crucial to prevent the development of gut lymphomas and other complications associated with gluten-induced enteropathy and malabsorption. Gluten is the main storage protein of wheat grains and is present in cereal species of the tribe Triticeae, which includes wheat, rye, and barley [60]. According with the U.S. Food and Drug Administration, foods that are labeled gluten-free must have level of gluten <20 ppm. Foods with these labels are naturally gluten-free foods, prepared foods that do not have a gluten-containing ingredient, and foods with a gluten-containing ingredient that has been processed to remove gluten. Processed foods must be prepared in a gluten-free ambient to avoid the cross-contamination. The main grains, starches, or flours that can be part of a gluten-free diet include amaranth, buckwheat, corn, flax, millet, quinoa, rice, sorghum, soy, and tapioca. Whereas in the past, oats were also considered not suitable for GFD, in the last years, some authors proved that oats of the tribe of Avenae can be consumed by celiac patients without risk, provided they are not contaminated by other cereals containing gluten [57]. GFD alleviates intestinal symptoms in an average of 3–6 months, much more quickly than dermatological ones. It can take even 1–2 years for the complete resolution of cutaneous lesions and itching, which tend to recur within 12 weeks after the reintroduction of gluten. Moreover, from a histopathologic point of view, the recovery is not immediate: IgA antibodies con persist at the dermal–epidermal junction (DEJ) after many years of a meticulous GFD [60]. Even though GFD is essential in CD to provide an effective cure and avoid complications, many celiac patients are often not diligent [56]. Actually, GFD needs accurate monitoring of all ingested foods, being time consuming and potentially having important repercussions on social life [61]. Moreover, gluten-free foodstuffs are not widely available and are more expensive than their counterparts. These issues may be the cause of a poor compliance to the diet and favor the persistence of gastrointestinal and cutaneous manifestations of CD [62]. Nutrition education is critical to improve patient adherence to GFD, especially towards celiac children and teenagers. Notably, GFD is also linked to possible adverse effects. Avoiding gluten-containing cereals and eating ultra-processed gluten-free products can lead to an unhealthy diet, one that is poor in fiber and rich in sugar and fat [63]. Some studies evaluating the nutritional composition of processed gluten-free products demonstrated a higher level of lipid, trans fats, and salt compared to their gluten-containing counterparts [64,65]. These circumstances may predispose celiac patients to chronic constipation, impaired glucose tolerance, and weight gain [66,67,68]. Additional studies have shown that many gluten-free foods are not enriched and may be deficient in several nutrients, including folate, iron, niacin, riboflavin, and thiamine; moreover, gluten-free foods often show lower average protein content across core food groups, especially pasta and breads [67]. For these reasons, it is important that celiac patients receive instructions for a balanced diet, not only aimed at avoiding gluten. It is essential that, after the diagnosis, the physician initiate an immediate referral to a dietitian with expertise in CD for nutritional assessment, diet education, meal planning, and assistance with the adaptation to the challenging gluten-free lifestyle [69].

#### 3.4.2. Pemphigus

This group of blistering diseases is characterized by intraepidermal acantholysis (detachment of keratinocytes) due to autoantibodies that attack desmoglein. This is a family of desmosomial cadherins that plays a role in the formation of desmosomes that join cells to one another. The main diseases belonging to this group are pemphigus vulgaris (PV), pemphigus foliaceus (PF), and paraneoplastic pemphigus (PNP) [70]. PNP is usually a severe form of pemphigus associated with hematologic neoplasms, e.g., B cell lymphoma, chronic lymphocytic leukemia, Castleman’s disease, Waldenstrom’s macroglobulinemia, and thymoma, but also sarcomas and other solid tumors. Unlike DH, these diseases can severely affect not only the skin, but also the mucous membranes, especially the oral mucosa and possibly the esophageal mucosa in the most severe forms, affecting the ability to swallow. However, mucosal involvement is usually absent in PF [71]. In addition, in all variants, epidermal detachment may be so significant to require management in a burn center. Prior to availability of corticosteroid therapy, which represents a mainstay in the treatment, pemphigus has a high fatality rate. For these reasons, clinicians have to consider multiple issues affecting nutrition in these diseases: (1) patient’s difficulty in feeding due to oral/gastrointestinal tract lesions; (2) increased catabolism caused by epidermal detachment, and potentially by an associated neoplasm; (3) hydroelectrolytic imbalance caused by fluid leakage through skin lesions; (4) the need of vitamin D and calcium supplementation to prevent secondary osteoporosis due to prolonged corticosteroid therapy; (5) a variety of dietary factors that have been proposed to play important roles in the onset, progression, exacerbation, and treatment of this disease [72].

Different scoring systems to assess the severity of the disease have been proposed [71]. For simplicity, when body surface area (BSA) involvement is greater than 30% or there are at least two mucous membranes affected, the disease can be defined as severe [71]. In these cases, aggressive nutritional support is needed to minimize the protein losses and to promote the healing of mucocutaneous lesions. A high-protein diet is recommended, with 2–3 g/kg body weight of protein daily in adults. A nasogastric feeding tube is required in most patients, as mucosal erosions impair oral feeding. In patients who are able to eat, discontinuous tube feeding supplements are administered, also during the night [72]. Nasogastric tube, and possibly a venous access, are also essential to correct fluid and electrolytes imbalances. The collaboration between dermatologist and dietician is mandatory in the management of the severe cases because poor nutritional status is often underestimated and an important cause of morbidity [73]. As mentioned above, it is recommended that vitamin D and calcium supplementation be used at initiation of glucocorticoid treatment to prevent secondary osteoporosis [74]. A negative correlation between the serum vitamin D level and the severity of disease have been reported in one study [75]. Yamamoto et al. showed that vitamin D3 can inhibit the expression of desmoglein 3 in the skin; therefore, vitamin D3 supplementation could have a role in the pharmacological treatment of PV [76], even if further studies are needed.

There is a wide discussion about the potential role of dietary factors in the induction or modification of the course of the disease. The onset of pemphigus often depends on the interaction between environmental factors and genetic background. Fogo selvagem, also known as endemic PF, is often proposed as a classical example of immune-mediated blistering disease that is supposed to be triggered by the bite of an insect (Simulium nigrimanum) in predisposed patients for genetic and environmental reasons [77]. There are drugs recognized as possible inducers or aggravating factors of bullous diseases, and on the basis of similarity of their chemical structure, some dietary factors have been suggested to have a similar mechanism of action [78]. Epidemiologic data of populations where pemphigus is endemic, such as Amazonian Brazil and India, further support the link between diet and pemphigus [79]. Dietary factors suspected of being inducers of pemphigus are known to contain thiol compounds, isothiocyanates, phycocyanins, phenols, or tannins [80]. Assumptions are primarily derived from case reports, epidemiologic studies, and in vitro studies [81]; therefore, additional investigations are needed to provide a better understanding and indications in the treatment.

#### 3.4.3. Bullous Pemphigoid

Bullous pemphigoid (BP) accounts for 80% of the subepidermal immunobullous cases, representing the most common autoimmune blistering disease [82]. This disease is caused by autoantibodies against molecules localized at the DEJ, and therefore keratinocytes detachment occurs at the basement membrane zone with the formation of tense bullae and intense pruritus. This disease usually affects the elderly and spares mucous membranes. There are no recognized associations between the onset of BP and dietary factors. Only a case report of a dyshidrosiform pemphigoid, a rare variant of BP affecting predominantly palmoplantar surfaces, being induced by high dose of nickel in the diet, has been reported [83]. The disease is completely resolved on a nickel-free diet. Association between gluten sensitivity and BP have been postulated, but there is no relevant evidence and no case reports of BP improved with a GFD [84]. Although usually less severe than pemphigus, BP can cause epidermal detachment of large portions of skin surface and can require high dose of systemic corticosteroids, leading to the metabolic and nutritional complications already listed in the chapter above. Notably, the patients are often elderly and have comorbidities that can further complicate the nutritional aspect, such as congestive heart failure or chronic kidney disease, which can require, for example, a restriction in fluids and/or proteins in the diet [81]. In the most severe cases, the collaboration with an experienced dietician and geriatrician is fundamental.

### 3.5. Vitiligo

Vitiligo is a hypopigmentation disorder affecting about 1% of the European and U.S. population [85]. It is clinically characterized by well-defined white patches of depigmentation that frequently appear in visible parts of the body, leading to embarrassment and lowering of QoL (quality of life) [86]. Its pathogenesis is complex and not yet fully understood, but a growing pool of data supports the idea of an immune dysregulation triggered by oxidative stress in patients with genetic predisposition [87]. Vitiligo therapies traditionally consist of corticosteroids, immunomodulators, and phototherapy, which can be effective in controlling the disease, but they do not always prove to be satisfactory for the patient [88]. Nutrition could be useful in bridging this gap, i.e., making traditional therapy more effective, safe, and satisfactory. Several studies and of different design have been conducted in recent years to investigate its role in vitiligo. The main nutrients that have been studied to play a role in the disease are discussed below.

Regarding gluten, two patients with vitiligo showed some degree of repigmentation after a gluten-free diet [89]. One patient, on oral dapsone therapy, the use of which is described in vitiligo, showed rapid repigmentation of the face [90,91]. A second patient, this time with celiac disease, showed complete repigmentation with the gluten-free diet, maintained at seven years after initiation of therapy [92]. Gluten elimination might be useful in patients with both vitiligo and celiac disease, but further and more robust studies are currently needed to consider it a treatment option.

Phenylalanine has a central regulatory role in melanin metabolism, hence its therapeutic potential, which has been investigated in several studies. Cormane et al. [93] found repigmentation in 94.7% of patients treated with UVA combined with l-phe 50 mg/kg/day. Two trials conducted by Siddiqui et al. [94] investigated the efficacy of phenylalanine combined with phototherapy in vitiligo. In the first study, oral phenylalanine combined with UVA was shown to be superior to phenylalanine alone. In the second, the combination of phenylalanine and UVA was shown to be superior to the combination of placebo and UVA [95]. An additional study found that the addition of topical phenylalanine to the combination of UVA and oral phenylalanine was superior to the combination alone [95]. Thirty-six patients treated with topical phenylalanine combined with NB-UVB showed superior repigmentation compared with 33 treated with clobetasol ointment alone [96]. Although there are few studies regarding phe associated with UVA and even fewer with narrow band UVB (NB-UVB), both topical and oral phenylalanine could be a potential combination therapy for phototherapy.

Zinc regulates works as a cofactor for superoxide dismutase, an antioxidant in the skin, and plays a key role in melanogenesis [97,98]. Shameer et al. [99] found superior improvement in patients with vitiligo treated with topical corticosteroid combined with oral zinc 440 mg daily, compared with those treated with topical corticosteroid alone. Zinc might be a good combination therapy, but data are scarce on this.

The relationship between vitiligo and fatty acids has not been thoroughly investigated. It appears that patients with vitiligo consume more saturated fatty acids and less polyunsaturated fat than unaffected patients. Increased fat intake has been associated with the onset of vitiligo [100]. The antioxidant alpha-lipoic acid has recently been tested in vitiligo, but in combination with phototherapy and other antioxidants, such as vitamins C and E, with superior benefit compared with placebo [101].

As far as vitamins are concerned, the most studied are vitamin B12, vitamin E, and vitamin D. Vitamin B12 and folic acid are involved in DNA repair and methylation. Studies are conflicting regarding vitamin B12 and folic acid supplementation in vitiligo. Serum vitamin B12 levels in patients were found to be sometimes abnormally low and sometimes normal compared with unaffected patients [102]. In one experimental study, patients treated with a combination of vitamin B12 and folic acid and sun exposure repigmented more than patients treated with sun exposure alone; in another study, vitamin B12 and folic acid combined with NB-UVB did not show greater repigmentation than phototherapy alone [103,104].

The immunomodulatory and antioxidant properties of vitamin C have been extensively documented. Studies of its supplementation in vitiligo are few. Some consider it contraindicated, due to its skin-lightening activity, but it seems that its antioxidant power outweighs the risk [105]. An Indian study performed on 188 patients found no difference in disease progression between those who took ascorbic acid and those who did not [106].

Vitamin E is a fat-soluble vitamin with antioxidant properties. Low plasma and tissue levels of vitamin E have been found in patients with vitiligo [107]. Evidence shows that it may increase the efficacy of NB-UVB therapy, as well as protect against the oxidative stress of psoralen and ultraviolet A radiation (PUVA) therapy, but further data are needed to confirm its potential [108,109].

Scarce data are also available regarding vitamin D. A 2015 pilot study suggests that high level of vitamin D may play a role in reducing disease activity [110].

In the impact that diet has on the course of vitiligo, so-called “functional foods”, have also been studied.

The anti-inflammatory, antioxidant, and antiatherogenic properties of green tea are mainly given by its most abundant and active component, epigallocatechin-3-gallate (EGCG). Its benefit in vitiligo has only been tested in mice, and therefore it needs to be confirmed in clinical trials in humans. EGCG delayed the time of onset, prevalence, and surface area of monobenzone-induced depigmentation in murine models [111].

*Ginkgo biloba* exerts multiple anti-inflammatory and antioxidants effects. The extract is produced from the ginkgo tree, and its use in Chinese medicine is well established. Clinical studies conducted on *Ginkgo biloba* in vitiligo patients are particularly interesting and promising. Parsad et al. [112] demonstrated that *Gingko biloba*, administered in the dose of 40 mg 3 times a day, is effective in halting disease progression. Ten of the patients recruited in the study also showed marked to complete repigmentation. A second study, conducted without any other ongoing vitiligo therapy, confirmed the efficacy of *Ginkgo biloba* through clinometric indexes. The authors reported a mean increase in Vitiligo Area Scoring Index (VASI) of 15% and a decrease in Vitiligo European Task Force (VEFT) of 0.36, meaning a 6% decrease in lesion size. No alterations in plasma coagulation parameters were recorded [113].

Curcumin is the main natural lipophilic polyphenol found in *Curcuma longa* and in other *Curcuma* spp. [114]. An in vitro study on perilesional keratinocytes obtained from vitiligo patients has demonstrated its significant antioxidant properties [115]. When administered as a topical (tetrahydrocurcuminoid) in combination with NB-UVB phototherapy, it has been shown to be effective in skin repigmentation (superior to the control, NB-UBV alone) [116].

*Polypodium leucotomos* (PL) is a species of fern that has been studied for use in various skin conditions such as atopic dermatitis, psoriasis, and vitiligo. In vitro, it showed photoprotective effects, both in oral and topical formulation [117]. In vivo, evidence shows that it may be a promising combination therapy. It enhances the effect of both UVA and NB-UVB phototherapy and PUVA [118]. Another interesting study regarding a PL derivative, Anapso, over 5 months of treatment showed efficacy in disease control in all participants [119].

Studies have been conducted on other “functional foods” in vitiligo therapy, e.g., botanicals and herbs. Capsaicin was tested in vitro, and it seemed promising, but clinical studies are lacking [120]. Carotenoids showed protective effects when combined with phototherapy [120]. Khellin, administered topically or orally, appears effective when combined with UVA therapy [121]. Topical use of Nigella saliva oil could be a successful adjunctive treatment to traditional therapies [122].

According to current scientific evidence, some of the substances examined may have some effectiveness, even in monotherapy, but the role of nutrition seems to be that of adjunctive therapy to traditional ones, especially phototherapy. From a pathogenetic point of view, the antioxidant properties meet the idea of oxidative stress as a key moment in the onset of vitiligo, but many aspects are still to be clarified. We suggest consolidating the results already obtained with further clinical studies, especially for the substances that seem most promising, such as *Ginkgo biloba* and phenylalanine, as well as to consider other substances with antioxidant power as new therapeutic possibilities. Further studies could also concern combinations of these foods or supplements not yet investigated, or the association with other traditional therapies, such as systemic immunomodulators. The evidence, however, supports the benefit of a varied and balanced diet and proper integration in vitiligo.

### 3.6. Alopecia Areata

Alopecia areata (AA) is a frequent, non-scarring type of hair loss caused by an immune response to the hair follicle [123]. Hair loss occurs in patches over portions of the body but can occasionally become severe and affect the entire body. Indeed, the clinical manifestations of AA are varied, ranging from well-defined patches of hair loss on the scalp to complete hair loss on the scalp (alopecia areata totalis), or even the whole scalp and body (alopecia areata universalis) [124].

Around 2% of people will have the ailment at some point in their lives [125]. It is believed to be an autoimmune illness that is heritable in some cases [126]. Hair loss in AA is thought to be caused by an autoimmune-mediated death of hair follicles as a result of a loss of immune privilege in the hair structure. A fundamental hypothesis for the pathophysiology of AA is that it is an autoimmune disorder caused by a breakdown of hair follicle immune privilege, in which various immunological systems fail to prevent cytotoxic immune attack on cells and antigens present at that location [127].

The hair follicle is a microorganism that exists in an immunological and hormonal milieu of its own. Throughout the anagen phase, the hair follicle epithelium creates a region of immune privilege, one that is characterized by the reduced expression of major histocompatibility complex class Ia antigens and local synthesis of immunomodulating substances. The hair follicle privileged immunological status is critical for preventing autoreactive CD8+ T lymphocytes from recognizing anagen and melanogenesis-associated antigens. The breakdown of the processes that maintain the hair follicle immune privilege predisposes the hair follicle to inflammatory attack, which contributes to the pathogenesis of AA. There are accumulating data that underline the role of interferon gamma in causing immune privilege loss [128].

Patients may experience anxiety as a result of AA, which raises their chance of having psychological and mental problems [129,130].

AA has been linked to a variety of autoimmune diseases, bolstering the case for an underlying autoimmune etiology [131]. Additionally, atopic illnesses and mental health issues have been shown to be more prevalent in patients with AA [132].

Vitamin D’s involvement in keratinocyte proliferation and differentiation has been widely explored and proven in the past. Vitamin D is produced in keratinocytes from 7-dehydrocholesterol by ultraviolet B light (290–315 nm) or obtained through food and dietary supplements [133,134].

The role of the vitamin D receptor (VDR) in the hair cycle was suspected by the observation of alopecia universalis in a rare genetic disorder called type II vitamin D-dependent rickets (VDDR IIA) [135]. Patients with VDDR IIA have normal hair at birth, probably due to normal hair follicle morphogenesis, but their hair falls out between the ages of one and three months. Recent studies in mice and in vitro demonstrate that VDR plays a critical role in the postnatal maintenance of hair follicle. VDR expression increases throughout the late anagen and catagen phases, which is associated with decreased keratinocyte proliferation and greater differentiation, making the presence of VDR a need for maintaining a regular hair cycle [136]. However, the functions of vitamin D and the VDR in the hair cycle remain poorly understood, and therapeutic treatments for hair problems are lacking. Vitamin D, on the other hand, is a critical immunomodulator, and vitamin D insufficiency has been linked to a variety of autoimmune disorders [137]. Recent retrospective investigations comparing AA patients to controls indicate that patients had considerably lower vitamin D levels (in blood and tissue) [138,139], and levels inversely correlate with severity of AA [140,141,142].

Additionally, topical calcipotriol has been shown to be effective in the treatment of AA [143,144].

The rationale for the addition of vitamin D to AA is that it may have immunomodulatory effects on T cells and upregulate VDR expression in HF and epidermal keratinocytes. Vitamin D has long been known to inhibit the activity and development of T-helper 17 cells, downregulate T-helper 1 cells, and enhance the action of T-regs, resulting in immunomodulation.

Papadimitriou et al. described the effective treatment of three juvenile instances of alopecia totalis/universalis/focalis with oral calcitriol, its less calcemic analogue paricalcitol, and high-dose cholecalciferol. Within one month, one patient developed asymptomatic hypercalcemia–hypercalciuria and was transferred to an even higher equivalent paricalcitol dosage [145].

Narang et al. investigated the effectiveness of calcipotriol lotion at 0.005 percent twice daily for three months in 22 patients with AA and found a correlation between the result and blood vitamin D levels. Hair regrowth was detected in 59% of patients after 12 weeks of therapy, and response to treatment was improved in individuals with low vitamin D levels. As a result, the authors recommended that topical calcipotriol may be an alternate therapy for AA, particularly in vitamin D-deficient individuals [146].

Kim et al. documented another pediatric instance. After three months of treatment with calcipotriol solution (Daivonex, 50 g/mL) once daily, full regrowth was noted in the afflicted region [147].

The efficacy and safety of this therapy method are unknown; further research is needed to determine the appropriate form and dose of vitamin D administration, either alone or in conjunction with other medications, particularly in the juvenile population.

The growing popularity of anti-inflammatory diets has sparked interest in their possible therapeutic value in lowering inflammation associated with autoimmune diseases, which are covered in AA.

According to preliminary findings, a gluten-free diet may be a helpful therapy for AA patients who also have celiac disease, a well-known comorbidity [148,149,150].

In patients with inflammatory and autoimmune illnesses, diets rich in fresh vegetables or rich in phytochemicals substances, such as carotenoids and polyphenols, with anti-inflammatory and antioxidant properties are suggested [151].

Vitamins and trace minerals are micronutrients that, while required in tiny amounts, are necessary components of our diet. Micronutrients are required for the regular hair follicle cycle due to their function in cellular turnover, which occurs often in the rapidly dividing hair follicle [152]. Additionally, several micronutrients have been shown to lower oxidative stress, a factor increasingly suspected in the etiology of AA [153,154,155].

Zinc is a critical mineral that is required for the catalytic activity of hundreds of enzymes. Zinc deficiency can cause significant changes in the hair, including telogen effluvium (TE) and the induction of fine, brittle hair [156].

However, clinical trials evaluating oral zinc as a therapy for AA have had conflicting results [156,157,158].

The only double-blind, placebo-controlled trial concluded that supplementation was not beneficial for patients with AA [159]. Iron insufficiency continues to be the most prevalent nutritional deficit worldwide, with persistent widespread telogen hair loss as a symptom. However, there is insufficient data in AA to urge iron deficiency screening. One limitation of studies on iron insufficiency is the predominance of female patients [160,161].

Generally, research on individual micronutrients and their therapeutic effect in AA has been conducted on a small number of patients. It is difficult to evaluate the impact of a single micronutrient in a cohort of participants eating a variety of diverse diets, and hence supplementation is not currently supported in patients with AA.

As with other inflammatory disorders, metabolic syndrome has been implicated in the development of AA [162].

Nasimi et al. discovered that 20 of 50 patients (54%) with alopecia areata had metabolic syndrome, a considerably greater rate than the 13 (26%) in controls [163]. In contrast, the frequency of metabolic syndrome was not statistically different between AA patients and healthy people in the research of Abdollahimajd et al.; nonetheless, AA patients had higher triglyceride levels and lower HDL cholesterol levels [164].

Although these studies promote a healthy diet and moderate physical activity, the small number of people enrolled precludes concluding that the metabolic syndrome is a substantial comorbidity in patients with AA. In the future, it would be fascinating to conduct research only on children.

## 4. Conclusions

Several dietary approaches, including the addition of specific nutrients, have been proposed to play a role in the pathogenesis, management, and/or therapy of psoriasis, atopic dermatitis, suppurative hidradenitis, bullous diseases, vitiligo, and alopecia areata. In some cases, it may be helpful to simply avoid established triggers. In others, supplements and dietary alteration are worthy of consideration; however, further studies regarding dietary manipulation and the effect of dietary components on these skin diseases are needed to better understand and treat patients.

## Data Availability

Not applicable.

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
