# Peer review of "The Role of Nutrition in Immune-Mediated, Inflammatory Skin Disease: A Narrative Review"

_nutrients, 2022, doi:10.3390/nu14030591_

Round 1

Reviewer 1 Report

The authors present a narrative review not a systematic review.

Major

The search algorithm is weak. I do not understand why only the term nutrition is used and other terms like "diet" "dietary" or similar were not used. I recommend the authors to ask for the help of somebody with expertise in bibliographic search.

The searh is indeed not systematic as some articles related for example with psoriasis that match the inclusion criteria of the authors are not included. 

A systematic review must have tables in the body of the manuscript or as an annex as they are the results of the study. The tables need to include many fields like design, patients, level of evidence. I suggest the authors to revise some systematic reviews in order to structure theirs.

A systematic review has a result section, a discussion and a conclussion.  In this paper everything  is mixed resulting in a narrative review. 

Author Response

Thanks for the suggestion. A revision of the article was made and the methods used were reviewed. The conclusion was that it is more accurate to call this type of manuscript a "narrative review." Therefore, the materials and methods were reviewed. The keyword problem was clarified in the materials and methods section. All text changes are highlighted in yellow.

Reviewer 2 Report

The manuscript by Diotallevi et al provides an interesting systematic review about the effects of nutrition on immune-mediated inflammatory skin diseases, which could be an important source for dermatologists and dieticians to find summarized information on this subject. The article provides sufficient background and is well organized, however some details have to be pointed out:

Throughout the manuscript, there are big blocks of text where the citation only appears in the beggining or in the end of the said block. Sometimes it is not easy to understand which part of the text is supported by each citation. Furthermore, the final references list is not uniform, i.e different formatting styles are used along the list, and they are not formatted acording to Nutrients's guidelines. Please correct. 

There is also reference to abbreviations and acronyms like SCORAD or NB-UVB without explaining their meaning the first time they appear.

The names of species and genera must be in italic

In lines 153-161 the authors point ou the position of two institutions regarding testing of serological markers of gluten sensitivity on psoriasis patients. However, it is not clear why these positions are stated. The authors should clarify.

Lines 201-203: The sentence starts and ends with "Predisposing factors for food restriction" which is very confusing. Please reformulate

Line 219: "According to some data obtained from literature reviews [35]" the authors use the plural but only cite one work.

Line 277: "Although basic scientific research supports a mechanistic role for flavonoids in in-275 flammatory disease," this portion needs reference.

Line 483: The number for this section is repeated. It should be 3.4.3.

Line 522 and 534: the word phenilalanine is spelled wrong. please correct

Lines 569-573: the authors state that only in vitro tests were performed, but then refer to murine models, which are in vivo. Please correct.

Line 695: it would be interesting if the authors added why "diets rich in fresh vegetables and a variety of protein sources are suggested".

Finally, I have detected various spelling and grammar errors along the manuscript. I recommend a carefull revision of the whole text, to correct these errors.

Author Response

Q: Throughout the manuscript, there are big blocks of text where the citation only appears in the beginning or in the end of the said block. Sometimes it is not easy to understand which part of the text is supported by each citation. Furthermore, the final references list is not uniform, i.e different formatting styles are used along the list, and they are not formatted according to Nutrients's guidelines. Please correct.

A: Many thanks for reviewing the text.The references have been revised. The style of the references has also been corrected. Corrections made are highlighted in yellow.

Q: There is also reference to abbreviations and acronyms like SCORAD or NB-UVB without explaining their meaning the first time they appear.

The names of species and genera must be in italic

A: Thanks for the suggestions. Changes have been made and highlighted in yellow.

Q: In lines 153-161 the authors point out the position of two institutions regarding testing of serological markers of gluten sensitivity on psoriasis patients. However, it is not clear why these positions are stated. The authors should clarify.

A: The positions of the organizations mentioned have been reported because these organizations are globally recognized for importance in the two diseases.

Q: Lines 201-203: The sentence starts and ends with "Predisposing factors for food restriction" which is very confusing. Please reformulate

A: Thanks for the suggestion. The sentence has been corrected to: Earlier AD onset, more severe disease and lower maternal education level seem to be the predisposing factors for food restriction practice.

Q: Line 219: "According to some data obtained from literature reviews [35]" the authors use the plural but only cite one work.

A: Thanks for the suggestion. The sentence has been corrected.

Q: Line 277: "Although basic scientific research supports a mechanistic role for flavonoids in in-275 flammatory disease," this portion needs reference.

A: Thanks for the suggestion. The reference was added.

Q: Line 483: The number for this section is repeated. It should be 3.4.3.

A: Thanks for the suggestion. Correction was made.

Q: Line 522 and 534: the word phenylalanine is spelled wrong. please correct

A: Thanks for the suggestion. Corrections was made

Q: Lines 569-573: the authors state that only in vitro tests were performed, but then refer to murine models, which are in vivo. Please correct.

A: Thanks for the suggestion.The term "in vitro" has been replaced with "in mice".

Q: Line 695: it would be interesting if the authors added why "diets rich in fresh vegetables are suggested".

A: To the sentence was added “rich in phytochemicals substances,  such as  carotenoids  and polyphenols,  with anti-inflammatory and antioxidant properties”

Q: Finally, I have detected various spelling and grammar errors along the manuscript. I recommend a carefull revision of the whole text, to correct these errors.

A: Thanks for the suggestions. A careful review of the grammar has been made.

Reviewer 3 Report

This is a very comprehensive review, dissecting the potential role of nutrients and the associated dietary-based lifestyle in affecting the onset and progression of a series of immune-related inflammatory skin disorders. The review also explores the potential therapeutic strategies that can be extrapolated from this systematic review. The manuscript is overall clear, well-written, and scientifically sound. The choice of references and the justifications of their critical appraisals made are appropriate. This review should appeal to the readers of this journal, based on the series of important novel correlations discussed related to dietary nutrition, regenerative medicine, and skin diseases.   

Author Response

Many thanks for reviewing the text and for the positive feedback.

Round 2

Reviewer 1 Report

No further comments.